# Wideband, Dual-Polarized Patch Antenna Array Fed by Novel, Differentially Fed Structure

Naiming Ou [1], Xian Wu [1,2], Kaijiang Xu [1], Fukun Sun [1,*], Tongfei Yu [1] and Yuchen Luan [3]

1   Department of Space Microwave Remote Sensing System, Aerospace Information Research Institute, Chinese Academy of Sciences, Beijing 100190, China; ounm@aircas.ac.cn (N.O.); wuxian22@mails.ucas.ac.cn (X.W.); xukj@aircas.ac.cn (K.X.); yutf@aircas.ac.cn (T.Y.)
2   School of Electronic, Electrical and Communication Engineering, University of Chinese Academy of Sciences, Beijing 100039, China
3   National Key Laboratory of Microwave Imaging Technology, Aerospace Information Research Institute, Chinese Academy of Sciences, Beijing 100190, China; luanyc@aircas.ac.cn
*   Correspondence: sunfk@aircas.ac.cn

**Abstract:** In this article, a $1 \times 4$ wideband, dual-polarized patch antenna array fed by a novel, differentially fed structure is proposed. The differentially fed structure of the antenna was realized by a parallel line structure that was printed on a PCB and connected with the inner and outer conductors of a coaxial cable. This method elaborately solved the problem of the narrow bandwidth of conventional microstrip differential feeding. By using a relatively thick air substrate (thickness = $0.19 \lambda_0$), stacked patches, a coupling feeding structure, and a differential feeding structure with the novel design, the element of the patch antenna array introduced below operated from 0.415 GHz to 0.707 GHz (achieving the 52.0% bandwidth) with a VSWR < 2.0, yielding a high port isolation less than $-28$ dB. For the array, an active VSWR less than 2.0 was also obtained with a port isolation of less than $-25$ dB, ranging from 0.405 GHz to 0.696 GHz. In the desired bandwidth, the array had an azimuth 3 dB beamwidth of about $19°$ for both horizontal polarization and vertical polarization. The antenna array also had good performance in scanning (stable gain and 3 dB beamwidth) and circular polarization (a 3 dB axial ratio bandwidth better than 54.5%).

**Keywords:** patch antenna; differentially fed; wideband; dual polarized; low cross-polarization; circular polarization

## 1. Introduction

Owing to its outstanding advantages in structure and working performance due to, for example, its low profile, lightweightness, and easy-to-realize dual polarization or circular polarization, the microstrip patch antenna is frequently utilized in the synthetic aperture radar (SAR) system of microwave remote sensing [1–6]. It can not only meet the size and weight requirements of the system but also facilitate the acquisition of various polarization scattering information, which is beneficial for the classification and recognition of different targets. However, a narrow bandwidth is always an obstacle that limits certain applications of traditional microstrip patch antennas. For an SAR system, in order to achieve high resolution in both range and azimuth directions, wideband, frequency-modulated signals; pulse compression techniques; and azimuth synthetic aperture technology are adopted. Consequently, antennas with a wide bandwidth are required for a high-resolution SAR system. The relative bandwidth of a microstrip patch antenna is generally 1–10% for a substrate thickness less than $0.06 \lambda_0$ [7], which makes it difficult to meet the requirements of an SAR system for high resolution and good image quality. To overcome this problem, a variety of measures can be taken, such as increasing the thickness of the substrate [8–10], adding parasitic patches [8,9], opening a U-shaped slot on the patch [10], or improving the feeding network [1,5,11,12]. For improving feeding, aperture coupling and L-probes

are extensively utilized. Compared with L-probes, feeding through aperture coupling can achieve better cross-polarization performance. Despite all this, modified probes were still adopted for both SAR systems in [1,5] because the weight of the antenna must be considered when applied to airborne SAR systems. When aperture coupling is exploited for feeding, it is necessary to add an additional metal reflector under the aperture to avoid high back-lobe radiation, which greatly increases the weight of the antenna.

For the design of a dual-polarized antenna, low cross-polarization and high port isolation are two important indicators. In many designs, differential feeding has been used to meet these requirements [13–18]. For differential feeding, broadband design is a tricky question. As described in [13], the conventional differential feeding balun only provides a consistent 180° (±5°) phase shift over a narrow band of around 5%. In order to realize broadband feeding, [13] adopted a −3 dB Wilkinson power divider cascaded with a broadband 180° phase shifter. The bandwidth was broadened to a certain extent but was still limited. Another method is directly using two ports to feed [14], which fundamentally solves the problem of bandwidth limitation caused by the feeding network. But, unfortunately, it also brings a new technical problem. In these cases, extra measures are needed to ensure the synchronous transmission of differential signals at the same time.

In this article, a $1 \times 4$ wideband, dual-polarized patch antenna array fed by a novel, differentially fed structure is introduced. The differentially fed structure of the antenna was realized by a parallel line structure that was printed on a PCB and connected with the inner and outer conductors of a coaxial cable. Instead of using out-of-baluns or power dividers or using the method of directly utilizing two ports to feed, this method elaborately solves the problem of the narrow bandwidth of traditional microstrip differential feeding. To widen the bandwidth, the probes at the end of the network were also improved, which generated large capacitive coupling with the radiation patches. In general, the performance of the antenna was as follows:

(1) Wide bandwidth and low cross-polarization. The proposed $1 \times 4$ antenna operated from 0.405 GHz to 0.696 GHz in *P*-band (the bandwidth up to 52.9%), with a VSWR < 2.0 and a cross-polarization level of less than −25 dB.

(2) Good performance when scanning. Scanning capability is a major advantage for array antennae and for the antenna designed in this study. When scanning at different angles, the gains and 3 dB beamwidth of the radiation patterns were stable.

(3) Wideband circular polarization capabilities. By measuring the prototype, it was found that the proposed antenna also worked well in left-handed circular polarization (LHCP) and right-handed circular polarization (RHCP), with a 3 dB axial ratio (AR) bandwidth of more than 54.5%.

The remainder of this article is organized as follows. In Section 2, the configuration of the proposed antenna is introduced. Section 3 provides the results obtained by simulation or experiment. These results exhibit a satisfying capability of the antenna, including properties of wide bandwidth, dual polarization, circular polarization, and scanning. Finally, Section 4 concludes this article.

## 2. Antenna Configuration

The configuration of the proposed $1 \times 4$ patch antenna array is presented in Figure 1. The size of the array was 1400 mm $\times$ 425 mm $\times$ 101 mm (height = 0.19 $\lambda_0$, where $\lambda_0$ represents the free-space wavelength at 550 MHz), with a spacing of 325 mm (0.60 $\lambda_0$) between adjacent elements. The frame in blue shows the element of the array, which was mainly composed of a ground, a feeding structure, and two stacked radiation patches. In order to achieve the same resonant frequencies for both linear polarizations, the shape of the stacked radiation patches was square [19]. To extend the operating bandwidth, the stacked patches configuration was adopted. The advantage of this configuration is that it could not only significantly broaden the bandwidth but also bring greater flexibility to the design. By adjusting the heights and sizes of the two patches, the characteristic parameters of the antenna, such as bandwidth, gain, efficiency, etc., could be optimized. After simulation

and optimization, the heights of the upper and lower patches were determined to be h1 = 24.0 mm and h2 = 71.0 mm, respectively, with patches having a thickness of 2.0 mm. The widths of the upper and lower patches were w1 = 169.5 mm and w2 = 207.0 mm, respectively. As shown in the figure, the patches were supported by eight polyimide columns (structure in blue, $\varepsilon_r$ = 3.5) and two circuit boards (structure in green, fabricated in Rogers RO4350B, $\varepsilon_r$ = 3.66). These two materials have similar dielectric constants and sufficient mechanical strength. The two structures ultimately provided support to the radiation patches through screw connections. To facilitate the installation of polyimide columns, corresponding mounting holes were reserved on the radiation patches and the metal ground. These mounting holes were all located on the diagonal of the radiation patches. The mounting holes for the upper columns had a diameter of 8.0 mm and were positioned 56.6 mm away from the center of the patches. Those for the lower had a diameter of 10.0 mm and were 91.9 mm away from the center of the patches. Based on this design, each part of the antenna could be manufactured separately and then conveniently assembled to form a whole.

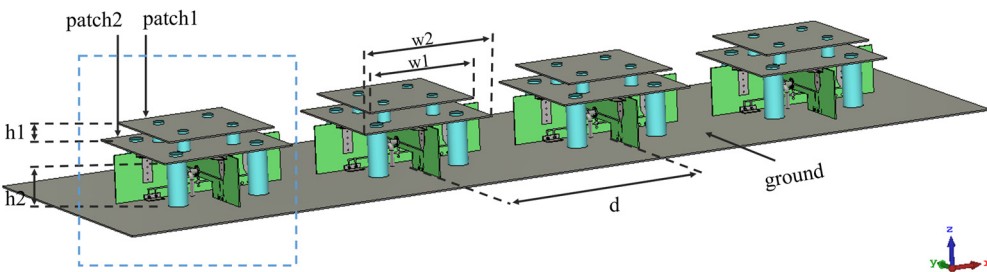

**Figure 1.** Configuration of the proposed 1 × 4 patch antenna array. (h1 = 24.0 mm, h2 = 71.0 mm, w1 = 169.5 mm, w2 = 207.0 mm, d = 325.0 mm).

The structure of the feeding network is shown in Figure 2. It was composed of two differentially fed structures printed on the PCB, which were placed orthogonally together as shown in Figure 2a. The thickness of the dielectric substrate was 1.52 mm. At this thickness, the narrowest stripline of the feeding network was 1.8 mm, which avoided increasing the manufacturing difficulty due to the feeding network being too narrow. Feeding through port 1 and port 2, dual polarization or other working conditions could be achieved. When the antenna array was applied to airborne SAR systems, the axis of the array followed the direction of the flight. For convenience, based on the distribution of the two polarization electric fields and the installation direction of the antenna during flight, we specified that the electric field generated by the excitation of port 1 was horizontal polarization (H-polarization). Similarly, port 2 was vertical polarization (V-polarization). Figure 2b,c show the details of the feeding network for H-polarization and V-polarization separately. Note that the differentially fed structure consisted of three components, including one coaxial cable (frame in red), one parallel line structure (frame in orange), and two probes (frame in blue). When the coaxial cable connected with the parallel line structure, the outer conductor was linked to the feeder on the same side, while the inner conductor connected with another feeder through the substrate. At the end of the parallel line structure, one side was connected to the probe and the other side was grounded. In this way, differential feeding and good impedance matching were achieved. It is also worth noting that the probes for the two polarizations had different lengths. The H-polarized probe was 50.5 mm, while the V-polarized probe was 33.7 mm. However, both polarizations had a parallel line structure of 162.0 mm. Therefore, to ensure that the electrical length of the feeding networks for the two polarizations was consistent, different lengths of coaxial cables were used for port 1 and port 2. The one connected to port 1 was 41.7 mm and the one connected to port 2 was 70.8 mm. The coaxial cables used were semi-rigid cables that had good performance in strength and stability.

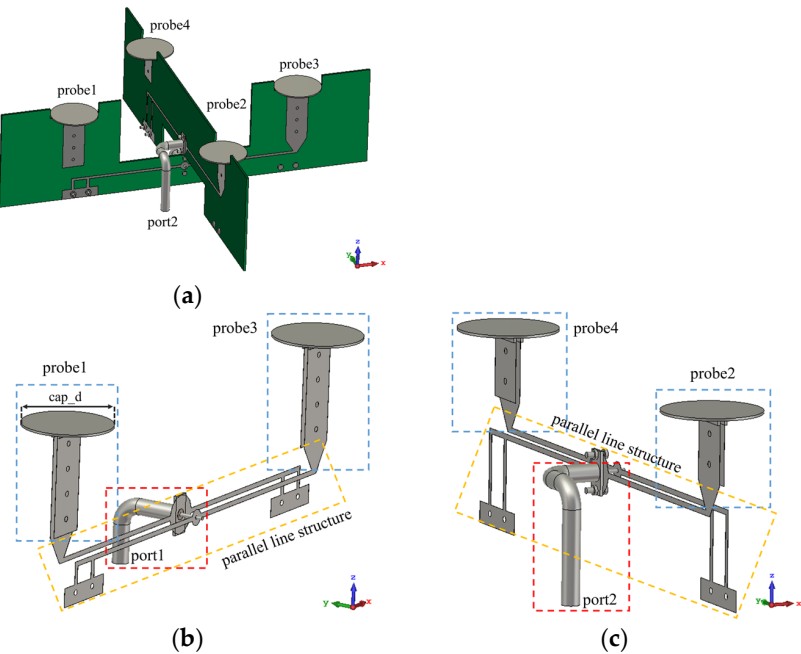

**Figure 2.** (**a**) The whole feeding network. Feeding network for (**b**) horizontal polarization and (**c**) vertical polarization. (For the convenience of observation, the parts connected to the inner conductor were moved 5 mm along the inner conductor in (**b**,**c**)).

The probes were not directly connected to the radiation patches and there was a 7 mm gap between them. Consequently, the patch was fed through coupling so that the bandwidth of the antenna was broadened. One more thing to say is that, unlike traditional L-shaped probes and the probe in [1], one part of the probe of the proposed antenna was a large circular cap with diameter cap_d = 33.1 mm, while the other part was printed on the PCB, so the feeding network provided large capacitive coupling with a simple structure and strong stability. In order to balance the current on the cap and avoid the distortion of the antenna pattern, a rectangular feeding structure connected with the cap on the backside of the PCB was connected with the feeding network through metalized vias. As a result, the antenna not only had good performance in bandwidth but also a simple structure and was easy to fabricate.

Thanks to the natural antiphase characteristic between the inner and outer conductors of the coaxial cable and the two conductors of the parallel line, the 180° phase difference between probes 1 and 3 or probes 2 and 4 was ingeniously realized. Figure 3 shows the distribution of the E-field when fed through port 1. The overall distribution of the E-field in the xoz-plane around the patches can be seen in Figure 3a. The phase of the electric field along the *z*-axis at 0.2 mm above the center of probes 1 and 3 is shown in Figure 3b. It should be noted that a consistent 180° (±2.6°) phase difference between probe 1 and probe 3 over a wideband of 57.5% was obtained, ranging from 0.390 GHz to 0.705 GHz, which indicated that differential feeding was achieved within a broad bandwidth.

Fed by this novel, differentially fed structure, the current distributions on the patches showed good regularity. Current distributions on the surface of the patches at 550 MHz are shown in Figure 4, where Figure 4a is H-polarization excited by port 1 and Figure 4b is V-polarization excited by port 2. For H-polarization, strong synclastic currents were generated along the two sides of the *x*-axis on the patches. Opposite currents were generated along the two sides of the *y*-axis. The currents were also opposite at the symmetrical position about the *x*-axis on the same side. By these means, the cross-polarization components (vertical polarization components) were effectively suppressed. Similar results were also observed when fed through port 2. That is, strong synclastic currents were generated along the two sides of the *y*-axis on the patches, while currents on the two sides of the *x*-axis

were inversely distributed about the *x*-axis and *y*-axis. Therefore, for vertical polarization, horizontal polarization components were also effectively suppressed.

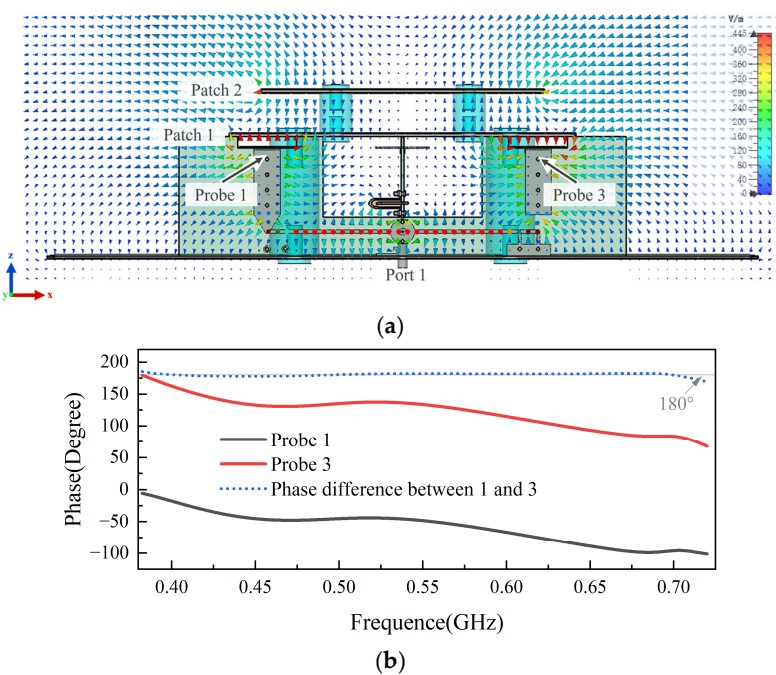

(**a**)

(**b**)

**Figure 3.** (**a**) Distribution of the E-field. (**b**) The phase of the E-field near probes 1 and 3.

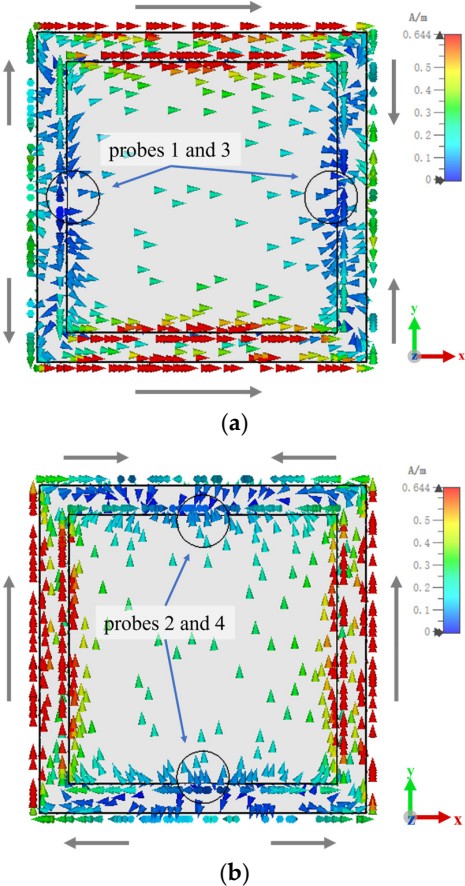

(**a**)

(**b**)

**Figure 4.** Current distributions on the patches. (**a**) H-polarization. (**b**) V-polarization.

More details about the proposed differentially fed structure are shown in Figure 5. The final optimized values of the corresponding parameters are given in Table 1.

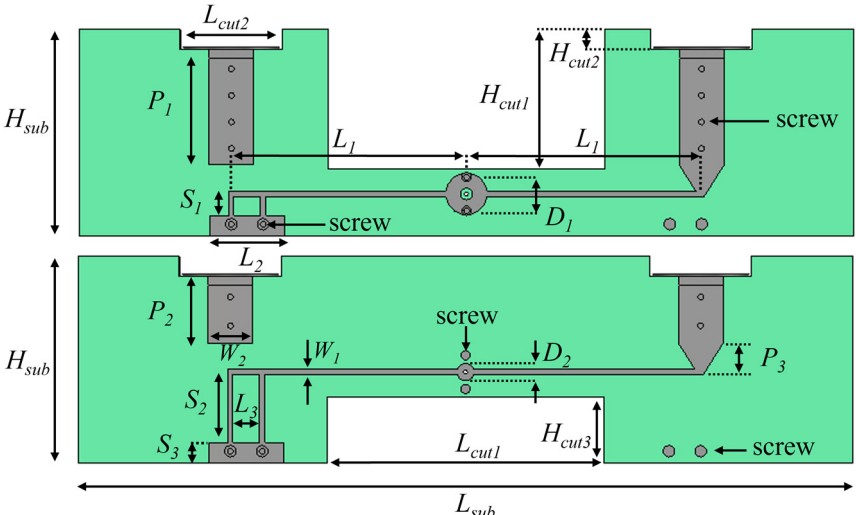

**Figure 5.** Design parameters of the feeding network.

**Table 1.** Antenna configurational parameters.

| Parameters | Value, mm | Parameters | Value, mm |
|---|---|---|---|
| $L_1$ | 81.0 | $S_1$ | 8.3 |
| $L_2$ | 26.0 | $S_2$ | 25.1 |
| $L_3$ | 9.1 | $S_3$ | 7.0 |
| $W_1$ | 1.8 | $H_{sub}$ | 71.0 |
| $W_2$ | 15.5 | $L_{sub}$ | 267.0 |
| $D_1$ | 14.4 | $L_{cut1}$ | 95.0 |
| $D_2$ | 6.0 | $L_{cut2}$ | 35.1 |
| $P_1$ | 39.6 | $H_{cut1}$ | 48.2 |
| $P_2$ | 22.8 | $H_{cut2}$ | 7.0 |
| $P_3$ | 10.9 | $H_{cut3}$ | 22.8 |

## 3. Results and Analysis

To validate the design, the proposed antenna was simulated with the electromagnetic simulation software CST Studio Suite 2022 [20] and verified by experiments. The results convincingly demonstrated that the antenna had good performance when it operated in *P*-band across the desired bandwidth.

### 3.1. Antenna Element

As described in Section 2, each antenna element had two ports to achieve dual polarization. More specifically, horizontal polarization was achieved through port 1 and vertical polarization was achieved through port 2. Simulated with the full-wave analysis software, the VSWRs and S-parameters of the antenna element were obtained and are shown in Figure 6. It can be observed that the impedance bandwidth for VSWR < 2.0 was over 52.0% for each port, ranging from 0.415 to 0.707 GHz. It can also be seen from the figure that S12 and S21 were lower than −28 dB in the bandwidth and even lower than −44 dB from 0.508 GHz to 0.707 GHz, which indicates that the antenna element possessed good port isolation characteristics and was conducive to the operation of the wideband, dual-polarized patch antenna.

Figure 7 shows the 3D far-field radiation patterns of the antenna element at five frequency points from 0.40 to 0.70 GHz. The two columns of patterns correspond to H-polarization and V-polarization, respectively. It can be seen from the figure that the radiation patterns were symmetrical within the working band for both polarizations. The

gains of the element were around 6.5 dBi to 9.0 dBi, which was related to the impedance matching and radiation efficiency of the corresponding frequency.

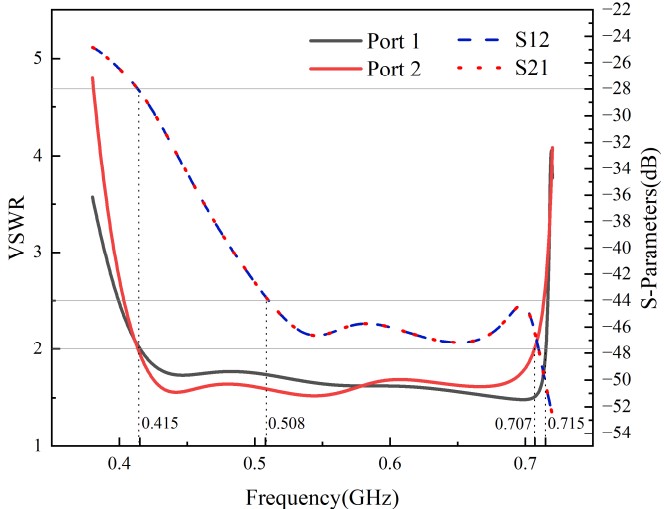

**Figure 6.** The simulated VSWRs and S-Parameters of the element.

Antenna radiation efficiency is an important parameter of antennas. The simulated efficiency of the proposed antenna element is shown in Table 2. Simulated through the software CST, the incident power and radiation power could be obtained. The values of the efficiency given in the table were obtained by dividing the radiation power by the incident power. As the conductor was assumed to be an ideal conductor and the losses were neglected when simulated, the results were slightly higher than the actual values. The difference in efficiency within the bandwidth was mainly caused by the VSWR and dielectric loss. As shown in Figure 6, the antenna element exhibited a relatively poor VSWR at 0.4 GHz. Consequently, the radiation efficiency of the antenna was also lower at this frequency. Overall, the radiation efficiency of the antenna was relatively high, ranging from 77.9% to 95.4%.

**Table 2.** The simulated efficiency of the proposed antenna element.

|  | 0.40 GHz | 0.50 GHz | 0.55 GHz | 0.6 GHz | 0.70 GHz |
|---|---|---|---|---|---|
| H-polarization | 80.1% | 91.8% | 93.8% | 94.7% | 86.3% |
| V-polarization | 77.9% | 92.7% | 95.4% | 95.4% | 86.0% |

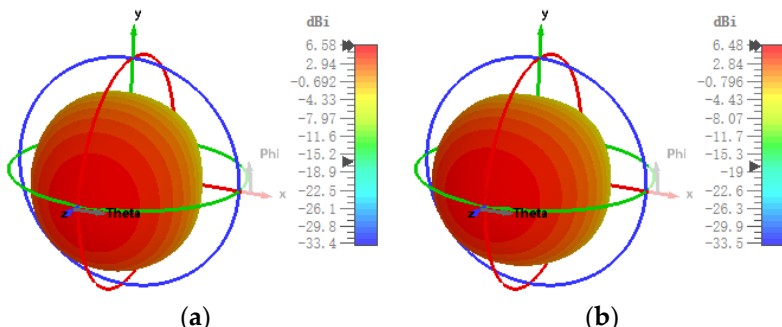

**(a)**          **(b)**

**Figure 7.** *Cont.*

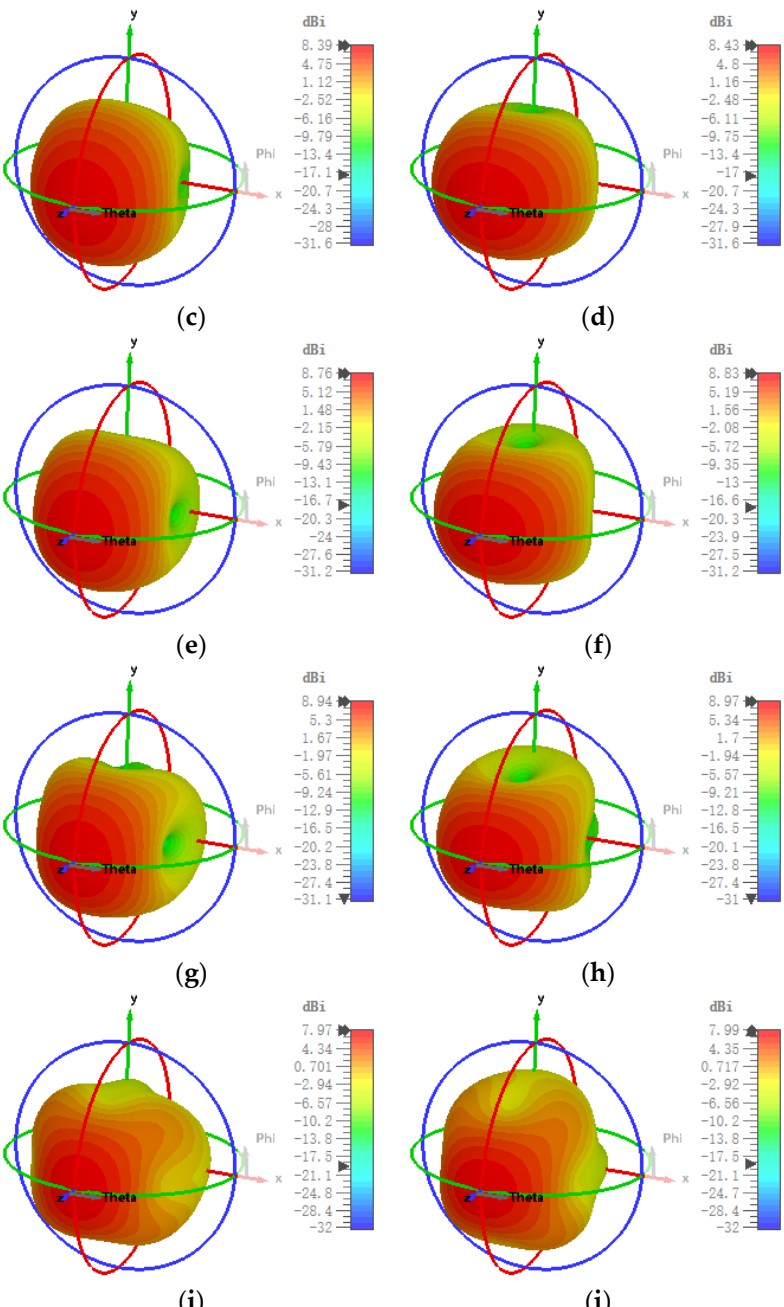

**Figure 7.** Simulated 3D far-field patterns in the desired band. (**a**) H-polarization at 0.40 GHz. (**b**) V-polarization at 0.40 GHz. (**c**) H-polarization at 0.50 GHz. (**d**) V-polarization at 0.50 GHz. (**e**) H-polarization at 0.55 GHz. (**f**) V-polarization at 0.55 GHz. (**g**) H-polarization at 0.60 GHz. (**h**) V-polarization at 0.60 GHz. (**i**) H-polarization at 0.70 GHz. (**j**) V-polarization at 0.70 GHz.

*3.2. Active VSWRs and Isolation of the 1 × 4 Antenna Array*

Figure 8 is the prototype of the designed 1 × 4 antenna array. Composed of the four elements described above, the array had eight feeding ports, numbered 1 to 8 in Figure 8a. Among them, H1, H2, H3, and H4 were ports for horizontal polarization and V5, V6, V7, and V8 were for vertical polarization. Figure 8a also shows the feeding network of the antenna. It can be seen here that the substrates were hollowed out to reduce the weight of the antenna. Figure 8b shows the completed antenna array and the support structure of the patches. Ultimately, in order to protect the antenna array, the antenna array was assembled in the radome as shown in Figure 8c.

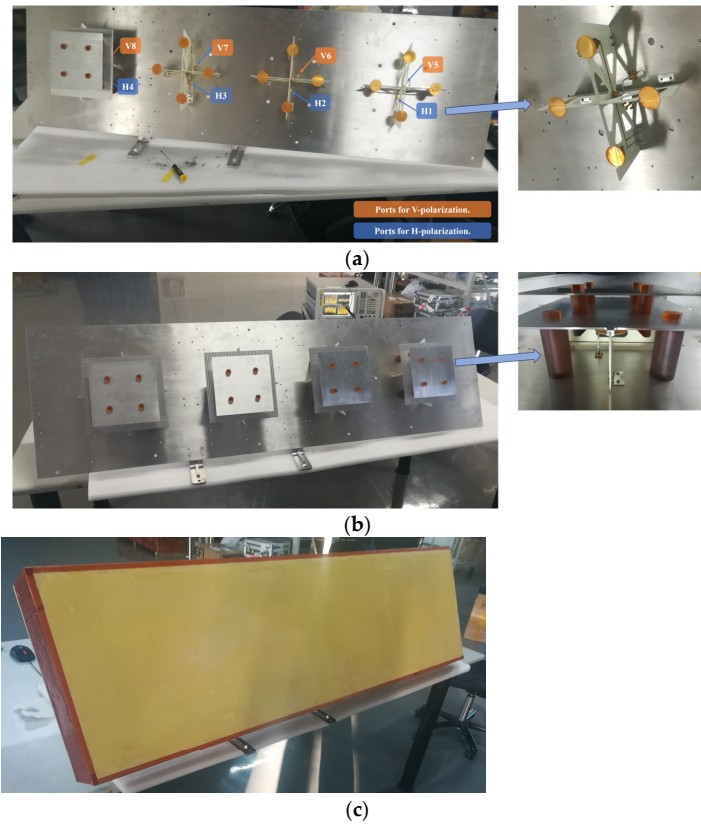

(a)

(b)

(c)

**Figure 8.** The prototype of the designed 1 × 4 antenna array. (**a**) Antenna array and the feeding ports. (**b**) Antenna array and the feeding network. (**c**) Antenna array assembled in the radome.

The active VSWRs and port isolation of the antenna array were studied and the corresponding results are displayed. As described in Figure 9, the active VSWR of each port was less than 2.0, except for port V6 and port V7 across the operational band from 0.392 to 0.696 GHz. The maximum VSWR for ports V6 and V7 was about 2.7. We hoped, though, that the values of the active VSWR of the full array for horizontal polarization and vertical polarization would be almost less than 2.0 in the wide bandwidth. The values were only slightly higher than 2.0. Figure 10 displays the port isolation of the array, from which we can see that port isolations lower than −25 dB were obtained, ranging from 0.405 to 0.707 GHz. Considering the active VSWR and port isolation, the proposed array exhibited an impedance bandwidth of 52.9% (i.e., 0.405–0.696 GHz).

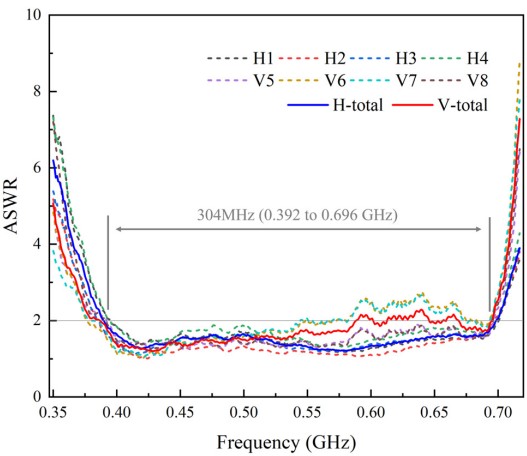

**Figure 9.** The active VSWRs of each port and array.

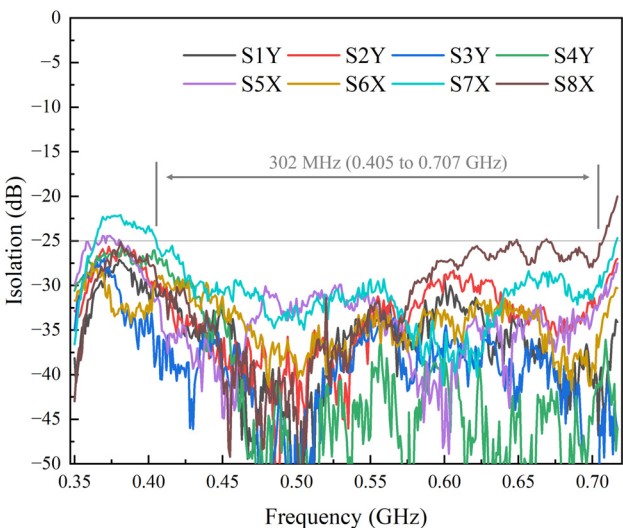

**Figure 10.** Port isolation of the antenna array.

### 3.3. Radiation Patterns of the Array

The radiation patterns and cross-polarization of the 1 × 4 antenna array were studied. The measured results are given in Figure 11. The four images describe the radiation patterns of the antenna with operating frequencies ranging from 0.40 to 0.70 GHz in different planes under different polarization states. For horizontal polarization, the 3 dB beamwidth in the azimuth direction was 19.6° ± 3.5°. For vertical polarization, it was 19.0° ± 3.1°. It could be also observed that the co-polarization was nearly 25 dB stronger than the cross-polarization in the boresight direction, which gives credit to the application of differential feeding and the design of the parallel line in the novel, differentially fed structure.

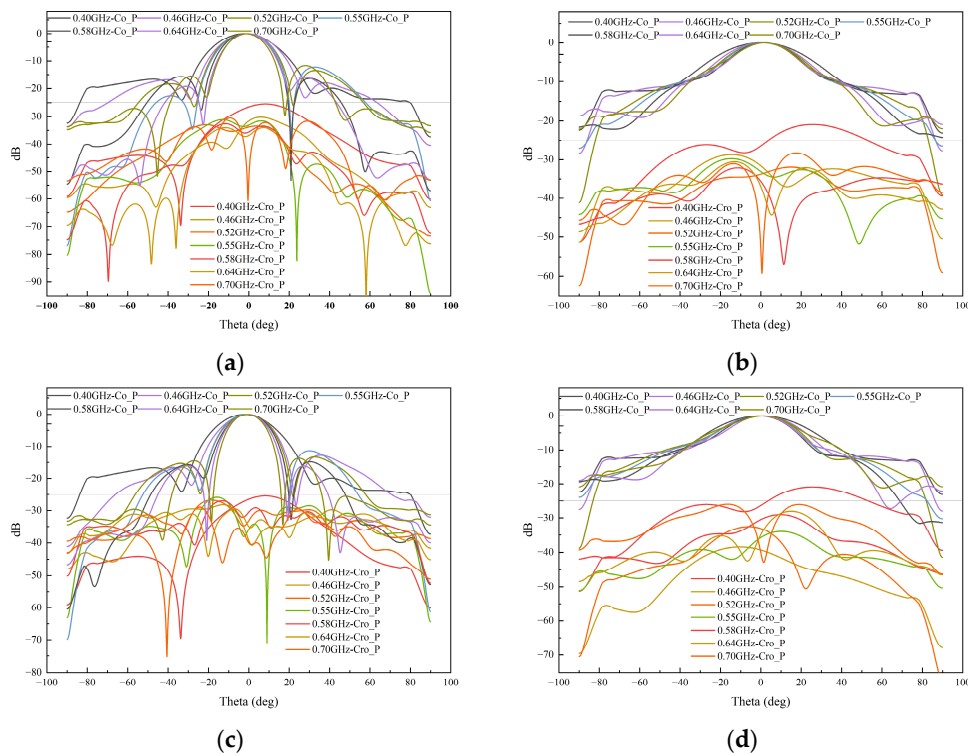

**Figure 11.** Measured radiation patterns of the proposed 1 × 4 antenna array of (**a**) horizontal polarization in the azimuth plane, (**b**) horizontal polarization in the elevation plane, (**c**) vertical polarization in the azimuth plane, and (**d**) vertical polarization in the elevation plane.

### 3.4. Scanning Performance

For an antenna array, as we all know, one of its significant properties is that the beam direction can be changed by adjusting the phase difference between the array elements without mechanical scanning. Figure 12 parades the scanning performance of the antenna array within the working band at 0.40 GHz, 0.55 GHz, and 0.70 GHz. According to the distance ($0.60 \lambda_0$) between adjacent elements, it could be inferred that the theoretical scanning range of the antenna array was about $\pm 40°$. However, for the intended application, only a small scanning angle was required. Therefore, we only utilized three sets of delay cables for the scanning test. Considering the requirement of the application and the constraint of our testing conditions, radiation patterns scanning at $0°$, $\pm 6°$, and $\pm 12°$ were measured, utilizing the delay cables. After scanning at different angles, the radiation patterns of H-polarization and V-polarization were presented. It can be seen from the figure that the pattern's gains were about 10 dBi, 13 dBi, and 14 dBi when the antenna scanned at different angles at 0.40 GHz, 0.55 GHz, and 0.70 GHz, respectively. The 3 dB beamwidth was also stable in different directions at the same frequency. The antenna still had a wide impedance bandwidth when working in scanning mode. As shown in Figure 13, the antenna still had a bandwidth of nearly 305 MHz with the active VSWR < 2.0 when its beam direction deviated from the normal direction at different angles for both polarizations.

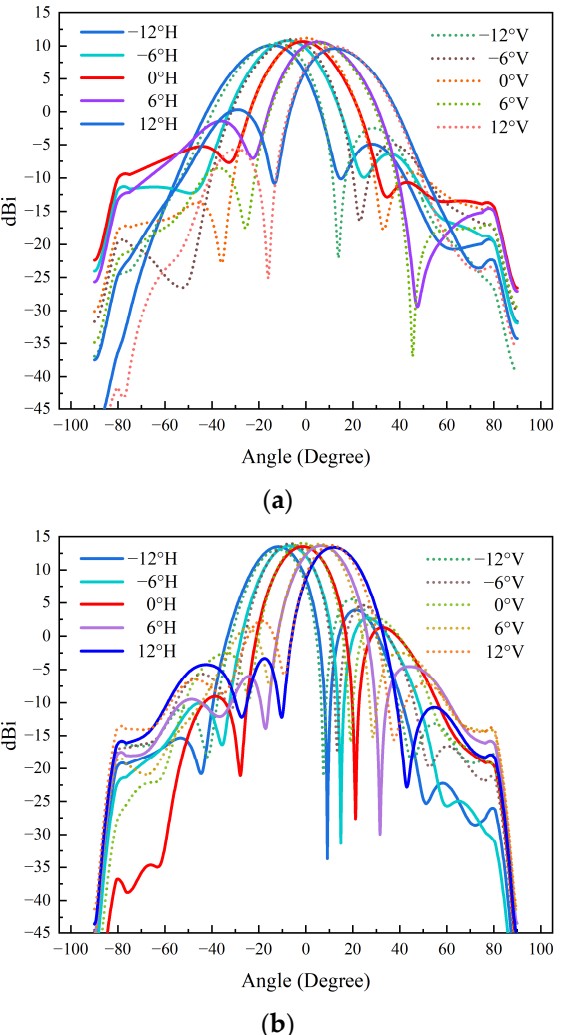

**Figure 12.** *Cont.*

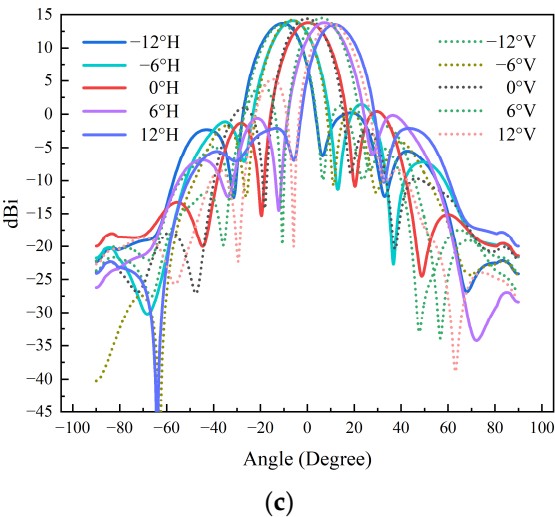

(**c**)

**Figure 12.** Measured radiation patterns when scanning at different angles at (**a**) 0.40 GHz, (**b**) 0.55 GHz, and (**c**) 0.70 GHz.

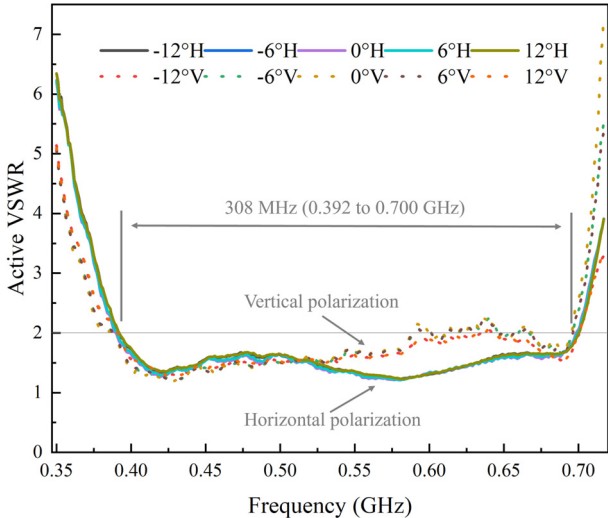

**Figure 13.** Measured active VSWRs of the array when scanning at different angles.

### 3.5. Circular Polarization

For the proposed dual-polarized antenna, by adjusting the phase difference between horizontal polarization and vertical polarization, left-handed circular polarization (LHCP) and right-handed circular polarization (RHCP) could also be realized, which is beneficial for radar to obtain more information. Specifically, when the phase difference between horizontal polarization and vertical polarization was 90° or −90°, circular polarization could be obtained. The measured axial ratio (AR) and maximum gain of LHCP and RHCP are given in Figure 14. According to the measured data, the 3 dB AR bandwidth was well above 54.5% (ranging from 0.4 to 0.7 GHz), with a maximum gain of more than 10.5 dBic. For circular polarization, the active VSWRs were also measured. Figure 15 shows that the impedance bandwidth (active VSWR < 2.0) was better than 300 MHz, both for LHCP and RHCP. In a nutshell, the 1 × 4 antenna array fed by the novel, differentially fed structure had competitive performance in circular polarization.

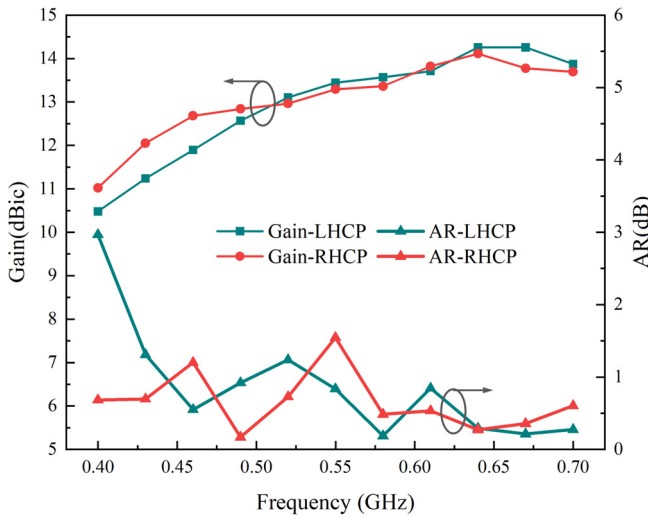

**Figure 14.** The measured AR and gain of the proposed antenna.

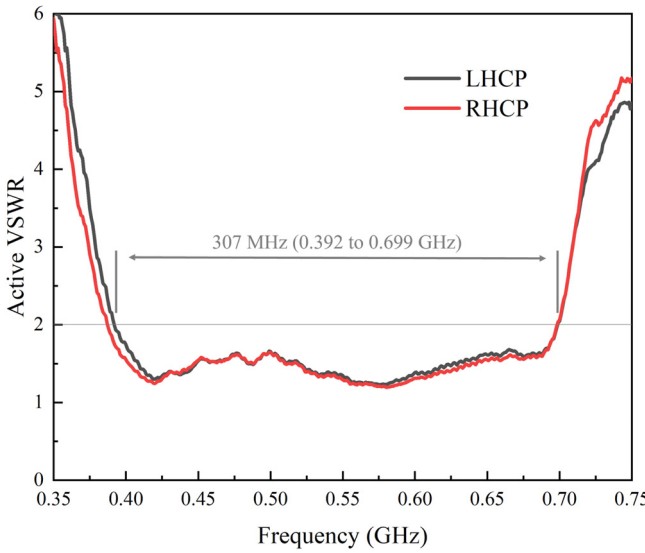

**Figure 15.** Measured antenna's active VSWRs of LHCP and RHCP.

## 4. Conclusions

In summary, a 1 × 4 wideband, dual-polarized patch antenna array fed by a novel, differentially fed structure that had good performance in the *P*-band was reported in this article. The novel, differentially fed structure elaborately realized differential feeding across a wide band and provided large capacitive coupling with the radiation patch with a simple and stable structure at the same time. As shown by the simulation data and experiment results, the proposed antenna possessed a wide impedance bandwidth and high port isolation. The 1 × 4 array not only had a large bandwidth of about 52.9% but also exhibited superior performance in scanning and circular polarization. It is worth noting that the use of polyimide columns to support the patches and the design of the feeding network also made the antenna simple in structure and easy to fabricate. Due to the characteristics above, together with the merits such as low profile and lightweightness, the proposed antenna array is a promising candidate for airborne SAR systems, especially for those that require multiple polarizations. SAR systems operating in the *P*-band compared to higher frequencies have superior penetration capabilities and are more sensitive to forest biomass. This antenna, combined with the characteristics of the *P*-band, can be applied to detect targets in areas covered by vegetation, ice, and snow or to measure forest biomass.

**Author Contributions:** Conceptualization, N.O., K.X., F.S. and T.Y.; validation, N.O., K.X., F.S. and T.Y.; writing—original draft preparation, X.W.; writing—review and editing, F.S., Y.L. and X.W.; supervision, N.O. All authors have read and agreed to the published version of the manuscript.

**Funding:** This research received no external funding.

**Data Availability Statement:** The data that support the findings of this study are available within the article.

**Conflicts of Interest:** The authors declare no conflicts of interest.

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
