# Peer review of "Wideband, Dual-Polarized Patch Antenna Array Fed by Novel, Differentially Fed Structure"

_electronics, doi:10.3390/electronics13071382_

Round 1
Reviewer 1 Report
Comments and Suggestions for Authors
20240322 Review Opinions:
Some questions:
l What simulation software was used?
l The simulated gain in Fig. 7 seems to be around 7-8 dB, but you stated in page 10, line 249 the gain is about 15 dB, and in Fig. 12 it seems the measured gain is higher than 15 dB, and in Fig. 14, for LHCP and RHCP, the gain is even higher, approaching 25 dB around 0.65 GHz, can you clarify these differences?
l What is the intended purpose for this SAR? The frequency seems a bit low to achieve enough special resolution, and scanning angle in Fig. 12 reaches only 12°?

Reviewer 2 Report
Comments and Suggestions for Authors
in this work the authors presented a 1×4 wideband dual-polarized patch antenna array fed by a novel differ entially-fed structure is proposed. some of my comments are
Fig. 1 can be improved with more clarity
Fig.8 a can be improved with different angle view
How much is the efficiency of the antenna
Comments on the Quality of English Language
English can be improved
